# Systematic Identification of Thiosemicarbazides for Inhibition of *Toxoplasma gondii* Growth In Vitro

**DOI:** 10.3390/molecules24030614

**Published:** 2019-02-10

**Authors:** Agata Paneth, Lidia Węglińska, Adrian Bekier, Edyta Stefaniszyn, Monika Wujec, Nazar Trotsko, Katarzyna Dzitko

**Affiliations:** 1Department of Organic Chemistry, Medical University, Chodźki 4a, 20-093 Lublin, Poland; lidia.weglinska@umlub.pl (L.W.); edyta.stefaniszyn@umlub.pl (E.S.); monika.wujec@umlub.pl (M.W.); nazar.trotsko@umlub.pl (N.T.); 2Department of Immunoparasitology, Faculty of Biology and Environmental Protection, University of Lodz, Banacha 12/16, 90-237 Lodz, Poland; adrian.bekier@unilodz.eu

**Keywords:** thiosemicarbazides, anti-*Toxoplasma gondii* activity, toxicity, SAR analysis

## Abstract

One of the key stages in the development of new therapies in the treatment of toxoplasmosis is the identification of new non-toxic small molecules with high specificity to *Toxoplasma gondii*. In the search for such structures, thiosemicarbazide-based compounds have emerged as a novel and promising leads. Here, a series of imidazole-thiosemicarbazides with suitable properties for CNS penetration was evaluated to determine the structural requirements needed for potent anti-*Toxoplasma gondii* activity. The best 4-arylthiosemicarbazides **3** and **4** showed much higher potency when compared to sulfadiazine at concentrations that are non-toxic to the host cells, indicating a high selectivity of their anti-toxoplasma activity.

## 1. Introduction

According to the World Health Organization, approximately up to one third of the world’s population is infected with *Toxoplasma gondii* (*T. gondii*). This protozoan is an obligate intracellular parasite with a complex life cycle which requires intermediate and definitive hosts. Intermediate hosts are especially all warm-blooded animals including most livestock, and humans. While *T. gondii* infect intermediate hosts and asexually reproduce in them, the only known definitive hosts in which this parasite may complete life cycle and sexually reproduce are members of the family *Felidae*. Cats become infected mainly through predation of intermediate hosts with latent parasite invasion. The parasite’s sexual replication takes place in the intestines and results in formation of oocysts. Cats shed large numbers of unsporulated oocysts with feces, for 1–3 weeks. In the environment oocysts sporulate within 1–5 days and become infective [1,2].

In *T. gondii* life cycle we can identify three main infective stages: tachyzoites, tissue cysts with bradyzoites and above-mentioned mature oocyst containing sporozoites. The main rout of parasite transmission to humans involves ingestion of either raw or underprepared meat containing tissue cysts or water or vegetables contaminated with soil containing oocysts. Additionally, people can become infected horizontally (iatrogenic) via blood transfusion or organ transplantation and vertically from mother to fetus via placenta. The parasite is also responsible for livestock infections. Farm animals, also these bred for human consumption, can acquire *T. gondii* infection through ingestion of sporulated oocysts with water or plants. After the release from tissue cysts and oocysts, which takes place in the intestines, sporozoites and bradyzoites, respectively, transform into the rapidly dividing tachyzoite (from *tachos*—speed in Greek) that is responsible for acute toxoplasmosis. In the immunocompetent individuals tachyzoites under the pressure of immune component convert into slow-dividing bradyzoites (*brady*—slow in Greek) enclosed within tissue cysts localized in various tissues e.g., neural or/and muscle. In people with a fully effective immune response those tissue cysts do not possess a direct threat to health and even life [2]. In contrast, in immunocompromised patients the rupture of tissue cysts leads to the release of bradyzoites. Their transformation to tachyzoites, in the absence of: nitric oxide, INF-γ, TNF-α, T cells and IL-12, results in disease reactivation. The process of tachyzoite–bradyzoite conversion is central to the pathogenesis and longevity of infection [3,4]. Therefore, the biggest challenge in the treatment of toxoplasmosis is related with acute phase, when rapidly multiplying tachyzoites are responsible for the numerous of necrotic changes and destruction of the host cells.

As mentioned earlier, although most human infections resolve without complications, they can be fatal or lead to serious problems in fetuses and immunocompromised patients [5,6,7,8]. Active *T. gondii* infection in HIV patients or those submitted for to cancer chemotherapy or organ transplants primarily leads to encephalitis, pneumonia or chorioretinitis but tissue destruction in other organs may occur as well [9,10]. Besides, when *T. gondii* primary infection is acquired during pregnancy, the vertical transmission may occur, resulting in birth defects such as hydrocephalus, epilepsy and mental retardation or even neonatal death [11,12].

Current first-line therapy for toxoplasmosis relies on inhibition of the folate pathway in the parasite, although antibacterial drugs have also been used with some success [13,14,15]. The most commonly used treatment is a combination of sulfonamides with 2,4-diaminopyrimidines, i.e., sulfadiazine with pyrimethamine—the principal drug combination or sulfamethoxazole with trimethoprim—alternate first line therapy [15,16]. These combinations are highly synergistic as the sulfonamide component inhibits dihydropteroate synthase—essential key enzyme for the utilization by the microorganism of 4-aminobenzoic acid in vital biosynthesis of dihydropteroic acid while 2,4-diaminopyrimidine component blocks dihydrofolate reductase—the enzyme essential for the conversion of dihydropteroic acid to tetrahydrofolate. Collectively, these components inhibit the parasite growth by blocking the biosynthesis of tetrahydrofolate, an essential factor needed for the production of nucleic acids which are required for DNA synthesis [17,18,19,20,21,22,23].

The combination of the sulfonamide with pyrimethamine, however, is highly effective in blocking replication of tachyzoites but has not activity on the latent bradyzoite form and therefore does not eliminate chronic infection [23,24,25]. Furthermore, pyrimethamine is associated with significant adverse reactions including anemia due to bone marrow suppression that requires coadministration of leucovorin [26,27,28]. As well, many patients experience intolerance or allergic reaction to the sulfa component [24]. Additionally, this therapeutic regiment requires long dosing periods and is contraindicated during the first two trimesters of pregnancy due to the potential for inducing development defects [16,29]. Other serious problems are the emergence of drug resistance and the incidence of relapses after discontinuation of therapy [25,30]. Although alternative drugs such as clindamycin, azithromycin have also been used to treat acute toxoplasmosis, they do not a clear chronic infection as well [14]. Atovaquone is also now in use and unlike to sulfonamide and pyrimethamine, this drug is effective against tachyzoite [31], besides other authors demonstrated that extended incubation of isolated brain cysts of *T. gondii* with atovaquone resulted in the inactivation of intracystic bradyzoite form [32,33]. Unfortunately, owing to the low-yield method for the synthesis and poor bioavailability, the cost of treatment with atovaquone is not affordable by patients in need, particularly in the third world countries [34].

Summarizing, currently, the only effective mean of avoiding *T. gondii* infection is a preventive healthcare, especially raising the awareness of future mothers and early diagnosis of pregnant women, and new-borns. An efficient method of complete elimination of the parasite from an infected organism has not yet been developed, so new agents or combinations of agents with greater therapeutic efficacy are necessary. Also, develop of safe and efficient tools for immunoprophylaxis of toxoplasmosis is still needed. Nowadays, only one vaccine containing live attenuated tachyzoites of *T. gondii* S48 strain, are available, but the potential use of the vaccine is restricted to the veterinary purposes because of the possible reversion of the attenuated mutant to the virulent strain [35].

One of the key steps in developing new therapies for the treatment of toxoplasmosis is to identify new non-toxic small molecules with high specificity to *T. gondii* for therapeutic investigation. In the search for such structures, thiosemicarbazides have emerged as a novel and promising leads. We have recently [36] tested a small series of 4-aryl(alkyl)thiosemicarbazides and found that all of them at non-toxic concentrations for the host cells were more effective than sulfadiazine. Compound **1** (Figure 1, left), the best of this group, was at least 15-fold more potent than sulfadiazine. Therefore, this class of compounds could offer a new way to combat the problem of infectious diseases induced by *T. gondii*. The conclusion from these studies was that the presence of five-membered heterocyclic moiety at the N1 terminal position and aryl moiety at the N4 terminal position of thiosemicarbazide core are the key functionalities required for potent anti-*T. gondii* activity. In this report, the systematic explorations of these initial findings are presented.

## 2. Results and Discussion

### 2.1. Rationale and Chemistry

As mentioned in the Introduction, in our preliminary study on the anti-*Toxoplasma gondii* activity of thiosemicarbazides we identified compound **1** (Figure 1, left) as the most potent in this respect [36]. The bioactivity of **1** was substantially lost when the thiophene ring was replaced with a similar in size furan or thiadiazole moiety. Although the thiosemicarbazides obtained were not enough to conduct an exhaustive SAR analysis, data reported there allowed us to make some preliminary observations regarding the effects of the type of substitution on bioactivity and other significant modifications. In particular, the obtained data highlighted that the need for the presence of a five-membered heterocyclic ring at the N1 terminal position and an aryl moiety at the N4 terminal position as the key functionalities required for potent anti-*Toxoplasma gondii* activity of thiosemicarbazides. Based on DFT studies, in turn, we have assumed that the steric hindrance between a substituent on the heterocyclic ring and the carbonyl group of thiosemicarbazide core is the factor reducing bioactivity.

With this information in our hand, a systematic SAR study was started with the aim of understanding better the substitution pattern at the N4 aryl moiety of azole-thiosemicarbazides on their anti-toxoplasma activity. As model compounds, series of 4-arylthiosemicarbazides with imidazole rings at the N1 position **2**–**7**, **10**–**22** was selected. Most of these compounds are close structural analogues of **1**–they possess similar molecular geometry (Figure 1, right) while the steric interactions between the methyl group of the imidazole ring and the carbonyl group of the thiosemicarbazide core were computationally shown to not be significant (Figure 1, middle). For comparison reasons, two imidazole-thiosemicarbazides with a benzyl group (compound **8**) or an alkyl chain (compound **9**) at the N4 position were also included in our SAR study.

As listed in Table 1, all designed imidazole-thiosemicarbazides **2**–**22** have suitable properties for CNS penetration; positive attributes include: (*i*) satisfaction of Lipinski’s rule of five (molecular mass less than 500 Da, log*P* not greater than 5, no more than 10 hydrogen bond acceptors, and no more than 5 hydrogen bond donors) [38], (*ii*) satisfaction of Veber’s descriptors (rotatable bond count not greater than 10 and polar surface area not greater than 140 Å^2^) [39], (*iii*) solubility expressed as log*S* within the range of −5 to −1 [40] and predicted absorption percentage (%ABS) within the range of 64.96 to 80.77 [41]. Thus, since designed imidazole-thiosemicarbazides **2**–**22** meet the requirements for good membrane permeability, all of them were subsequently prepared under routine protocol [42], in the one-step reaction of 4-methylimidazole-5-carbohydrazide with isothiocyanate (Scheme 1).

### 2.2. In Vitro Anti-Toxoplasma gondii Activity of the Imidazole-Thiosemicarbazides **2**–**22**

The first step of structure-activity relationship (SAR) study was to confirm the impact of the N4 aryl moiety and its substitution pattern on anti-*Toxoplasma gondii* activity of imidazole-thiosemicarbazides **2**–**22**. For this purpose, compounds **2**–**4** with an electron withdrawing nitro, and electron donating methoxy (**5**–**7**), benzyl (**8**), and 2-chloroethyl (**9**) substituents were prepared and tested. As presented in Figure 2, compound **9** with an aliphatic chain had only weak activity (IC_50_ > 125 µg/mL) thereby confirming our initial assumption regarding the favorable role of the N4 phenyl ring on activity. When the IC_50_ values of the remaining compounds were analyzed it was clear that the additional methylene bridge between the thiocarboxamide group and the phenyl scaffold had a negative effect on anti-*Toxoplasma* activity as well. Indeed, all tested 4-arylthiosemicarbazides **2**–**7** were more potent than the benzyl analogue **8** (IC_50_ ~ 149.59 µg/mL). Among them, compounds having strong electron-withdrawing *meta*-nitro group (**3**, IC_50_ = 14.78 µg/mL) or a *para*-nitro group (**4**, IC_50_ = 21.62 µg/mL) showed higher activity than those with strong electron donating *meta*-methoxy (**6**, IC_50_ = 79.65 µg/mL) and *para*-methoxy substitution (**7**, IC_50_ = 113.99 µg/mL). This might be either due to the differential uptake and distribution of the compounds into the tachyzoites’ cells or due to electronic effects of the substituents. In turn, the substitution pattern on the phenyl ring appears not to play a crucial role. Indeed, whilst within nitro series **2**–**4** isomers *meta*-**3** and *para*-**4** were the most potent, the best activity for methoxy series **5**–**7** was detected for the *ortho***-5** isomer (IC_50_ = 33.11 µg/mL).

With the aim of synthesizing new analogues that would significantly contribute to our envisaged SAR study, compounds **10**–**15** were synthesized and tested. As illustrated in Figure 3, the introduction of extra electron donating methoxy groups in the *meta* or *para* positions of weak inhibitors **6** or **7** leads to a further reduction in activity (**13**, IC_50_ ~ 174.98 µg/mL), thereby confirming our initial assumption that *meta* and *para* positions are not tolerated for substituents with electron donating character. Contrary to expectations, however, replacement of the electron withdrawing *meta-*chloro group of **10** (IC_50_ = 25.70 µg/mL) by small in size electron withdrawing *meta-*fluoro group (**11**, IC_50_ ~ 117.92 µg/mL) or a bulky electron withdrawing *meta*-acetyl group (**12**, IC_50_ = 95.50 µg/mL) reduced the activity while adding an extra *para* electron donating methyl group to the structure of **6** increased its potency significantly (compound **14**, IC_50_ = 18.62 µg/mL). The results of the bioassay for **15** were also not unambiguous. Indeed, the introduction of extra fluorine into the *ortho* position of **11** leads to increased activity, despite the fact that this position seemed to be unfavourable for substituents with electron withdrawing character as noted for the nitro series **2**–**4**.

Since the influence of chemical substitution and electronic effects on anti-*Toxoplasma gondii* activity of studied 4-arylthiosemicarbazides seems to be more complicated than expected, we also prepared a series of methyl isomers **16**–**18**, three compounds with *para*-substitution, i.e., *N,N*-dimethylamino **19**, *N,N*-diethylamino **20**, ethoxy **21**, and compound **22** with two isopropyl groups at the *ortho* position. According to the results presented in Figure 4, in the 4-arylthiosemicarbazide structure there is sufficient space at the *para* position for the chain extension, most likely for its potential interaction with the molecular target(s). Indeed, in contrast to weak inhibitor **22** (IC_50_ = 97.72 µg/mL) with two bulky isopropyl groups at the *ortho* position, compounds with *para* substitution **20** (IC_50_ = 25.70 µg/mL) and **21** (IC_50_ = 95.28 µg/mL) displayed improved *T. gondii* inhibition compared to their shorter by a methylene linker precursors **19** (IC_50_ ~ 164.70 µg/mL) and **7** (IC_50_ = 113.99 µg/mL). Surprisingly, similarly to the nitro series **2**–**4**, within the electron donating methyl series **16**-**18** the lowest antiparasitic effect was also noted for the *ortho* isomer **16** (IC_50_ ~ 150.31 µg/mL). The antiparasitic activity of the two remaining isomers, i.e., *meta*-**17** (IC_50_ = 37.15 µg/mL) and *para*-**18** (IC_50_ ~ 57.58 µg/mL), was approximately two times lower compared to nitro analogs **3** and **4**, respectively.

Summing up, SAR analysis indicates that steric theory alone, i.e., the interaction between thiosemicarbazide and molecular target(s), cannot fully describe the observed trend in bioactivity and that the non-correlation between the results of bioassay and substitution pattern at the N4 aryl moiety of imidazole-thiosemicarbazide is likely due to the combination of both steric and specific, i.e., hydrophobic and electrostatic, intermolecular interactions. Evidently, further studies are needed to understand the molecular basis of the anti-toxoplasma activity of this class of compounds.

### 2.3. Cytotoxicity of the Imidazole-Thiosemicarbazides **2**–**22** against L929 Cells and Selectivity Index

To ensure the imidazole-thiosemicarbazides **2**–**22** were targeting the parasite and not the host cells, a cytotoxicity screen against mammalian L929 cells was performed. Cytotoxicity, expressed as CC_30_, was defined as the highest dilution of test samples that causes 30% or greater destruction of cells. In Table 2 the results expressed as the percent of viable cells ± standard deviation in the concentration range between 3.91 to 500 µg/mL and the CC_30_ values are presented. As seen, only 5 (**8**, **11**, **14**, **18**, **19**) of 21 imidazole-thiosemicarbazides have unfavorable cytotoxic concentration close to or lower than IC_50_ values.

Based on the toxicity against L929 cells, the selectivity index (SI) was also calculated as the logarithm of the ratio of the CC_30_ and the IC_50_ values. The selectivity index is a widely accepted parameter used to express a compound’s in vitro efficacy in the inhibition of *T. gondii* proliferation and only compounds with SIs higher than 0 could be considered as selective. The higher SI ratio, the theoretically more effective and safe a drug would be during in vivo treatment for toxoplasmosis.

As shown in Table 3, except for **8**, **11**, **18** and **19**, all compounds showed selective inhibition of the parasite, thus confirming their safety profile. Important to note, the best inhibitors of *Toxoplasma gondii* proliferation **3**, and **4** presented also the highest selective index, making them a good candidate for future in vivo studies.

## 3. Materials and Methods

### 3.1. Chemistry

All commercial reactants and solvents with the highest purity were purchased from either Sigma-Aldrich (St. Louis, MS, USA) or Alfa Aesar (Karlsruhe, Germany) and used without further purification. The melting points were determined on a Fischer-Johns block (Fisher Scientific, Schwerte, Germany) and are uncorrected. Elemental analyses were determined by an AMZ-CHX elemental analyzer (PG, Gdańsk, Poland) and are within ±0.4% of the theoretical values. ¹H-NMR spectra were recorded on an Avance 300 MHz) spectrometer (Bruker BioSpin GmbH, Rheinstetten, Germany). Analytical thin layer chromatography was performed with 60F_254_ silica gel plates (Merck, Darmstadt, Germany) and visualized by UV irradiation (254 nm). The physicochemical characterization of compounds **7** and **18** was presented in [43,44]. The structure of **4** was presented in our previous publication [45], however no physicochemical characterization data was reported.

### 3.2. General Procedure for Synthesis of the Imidazole-Thiosemicarbazides **2**–**22**

A solution of 4-methylimidazole-5-carbohydrazide (0.01 mol) and an equimolar amount of an isothiocyanate (0.01 mole) in anhydrous ethanol (25 mL) was heated under reflux for 10–30 min. After cooling, the solid formed was filtered off, dried and crystallized from ethanol.

*1-(4-Methylimidazol-5-oyl)-4-(2-nitrophenyl)thiosemicarbazide* (**2**). Yield: 84%. M.p. 215–217 °C. ^1^H-NMR (DMSO-d_6_) δ (ppm): 2.45 (s, 3H, CH_3_); 7.35–7.40 (m, 1H, 1×CH_ar_); 7.62–7.75 (m, 2H, 2×CH_ar_); 8.02–8.05 (m, 1H, 1×CH_ar_); 8.31–8.34 (m, 1H, 1×CH_ar_); 9.61, 10.03, 10.25; 12.41 (4s, 4H, 4×NH). Anal. Calcd. for C_12_H_12_N_6_O_3_S (320.33): C 44.99; H 3.78; N 26.24. Found: C 44.84; H 3.77; N 26.26.

*1-(4-Methylimidazol-5-oyl)-4-(3-nitrophenyl)thiosemicarbazide* (**3**). Yield: 80%. M.p. 210–212 °C. ^1^H-NMR (DMSO-d_6_) δ (ppm): 2.46 (s, 3H, CH_3_); 7.56–7.64 (m, 2H, 2×CH_ar_); 7.95–8.04 (m, 2H, 2×CH_ar_); 8.53 (s, 1H, 1×CH_ar_); 9.85, 9.98, 12.41 (3s, 4H, 4×NH). Anal. Calcd. for C_12_H_12_N_6_O_3_S (320.33): C 44.99; H 3.78; N 26.24. Found: C 44.83; H 3.79; N 26.32.

*1-(4-Methylimidazol-5-oyl)-4-(4-nitrophenyl)thiosemicarbazide* (**4**). Yield: 80%. M.p. 235–237 °C. ^1^H-NMR (DMSO-d_6_) δ (ppm): 2.46 (s, 3H, CH_3_); 7.63 (s, 1H, 1×CH_ar_); 7.91–7.96 (m, 2H, 2×CH_ar_); 8.16–8.21 (m, 2H, 2×CH_ar_); 9.98–10.24 (m, 3H, 3×NH); 12.36 (s, 1H, 1×NH). Anal. Calcd. for C_12_H_12_N_6_O_3_S (320.33): C 44.99; H 3.78; N 26.24. Found: C 44.82; H 3.77; N 26.33.

*4-(2-Methoxyphenyl)-1-(4-methylimidazol-5-oyl)thiosemicarbazide* (**5**)**.** Yield: 87%. M.p. 200–202 °C. ^1^H-NMR (DMSO-d_6_) δ (ppm): 2.46 (s, 3H, CH_3_); 3.73 (s, 3H, OCH_3_), 6.89–7.13 (m, 4H, 4×CH_ar_); 7.64 (s, 1H, 1×CH_ar_); 8.92–10.03 (m, 3H, 3×NH); 12.44 (s, 1H, 1×NH). Anal. Calcd. for C_13_H_15_N_5_O_2_S (305.36): C 51.13; H 4.95; N 22.94. Found C 51.47; H 4.96; N 22.98.

*4-(3-Methoxyphenyl)-1-(4-methylimidazol-5-oyl)thiosemicarbazide* (**6**)**.** Yield: 86%. M.p. 196–198 °C. ^1^H-NMR (DMSO-d_6_) δ (ppm): 2.45 (s, 3H, CH_3_); 3.72 (s, 3H, OCH_3_), 6.68–7.13 (m, 1H, 1×CH_ar_); 7.06–7.09 (m, 1H, 1×CH_ar_); 7.18–7.24 (m, 2H, 2×CH_ar_); 7.62 (s, 1H, 1×CH_ar_); 9.59–9.76 (m, 3H, 3×NH); 12.40 (s, 1H, 1×NH). Anal. Calcd. for C_13_H_15_N_5_O_2_S (305.36): C 51.13; H 4.95; N 22.94. Found C 51.35; H 4.96; N 22.89.

*4-Benzyl-1-(4-methylimidazol-5-oyl)thiosemicarbazide* (**8**)**.** Yield: 90%. M.p. 225–227 °C. ^1^H-NMR (DMSO-d_6_) δ (ppm): 2.44 (s, 3H, CH_3_); 4.70–4.71 (d, 2H, 1×CH_2_); 7.26–7.32 (m, 5H, 5×CH_ar_); 7.59 (s, 1H, 1×CH_ar_); 8.31, 9.26, 9.67, 12.36 (4s, 4H, 4×NH). Anal. Calcd. for C_13_H_15_N_5_OS (289.36): C 53.96; H 5.23; N 24.20. Found C 54.01; H 5.22; N 24.11.

*4-(2-chloroethylphenyl)-1-(4-methylimidazol-5-oyl)thiosemicarbazide* (**9**)**.** Yield: 96%. M.p. 175–177 °C. ^1^H-NMR (DMSO-d_6_) δ (ppm): 2.44 (s, 3H, CH_3_); 3.88–3.93 (m, 4H, 2×CH_2_); 7.72 (s, 1H, 1×CH_ar_); 10.66, 11.59 (2s, 4H, 4×NH). Anal. Calcd. for C_8_H_12_ClN_5_OS (261.73): C 36.71; H 4.62; N 26.76. Found C 36.59; H 4.61; N 26.81.

*4-(3-Chlorophenyl)-1-(4-methylimidazol-5-oyl)thiosemicarbazide* (**10**)**.** Yield: 78%. M.p. 203–205 °C. ^1^H-NMR (DMSO-d_6_) δ (ppm): 2.43 (s, 3H, CH_3_); 7.15–7.16 (d, 1H, 1×CH_ar_); 7.30–7.35 (t, 1H, 1×CH_ar_); 7.47–7.53 (m, 1H, 1×CH_ar_); 7.63 (s, 1H, 1xCH_ar_); 7.70 (s, 1H, 1×CH_ar_); 9.71–9.81 (m, 3H, 3×NH); 12.40 (s, 1H, 1×NH). Anal. Calcd. for C_12_H_12_ClN_5_OS (309.77): C 46.53; H 3.90; N 22.61. Found C 46.67; H 3.89; N 22.65.

*4-(3-Fluorophenyl)-1-(4-methylimidazol-5-oyl)thiosemicarbazide* (**11**)**.** Yield: 81%. M.p. 191–192 °C. ^1^H-NMR (DMSO-d_6_) δ (ppm): 2.43 (s, 3H, CH_3_); 6.91–6.97 (m, 1H, 1×CH_ar_); 7.31–7.34 (m, 2H, 2×CH_ar_); 7.53 (s, 1H, 1×CH_ar_); 7.63 (s, 1H, 1×CH_ar_); 9.70, 9.82, 9.86, 12.40 (4s, 4H, 4×NH). Anal. Calcd. for C_12_H_12_FN_5_OS (293.32): C 49.14; H 4.12; N 23.88. Found C 49.29; H 4.13; N 23.79.

*4-(3-Acetylphenyl)-1-(4-methylimidazol-5-oyl)thiosemicarbazide* (**12**)**.** Yield: 78%. M.p. 206–208 °C. ^1^H-NMR (DMSO-d_6_) δ (ppm): 2.46 (s, 3H, CH_3_); 2.57 (s, 3H, COCH_3_); 7.43–7.48 (t, 1H, 1×CH_ar_); 7.63 (s, 1H, 1×CH_ar_); 7.71–7.74 (m, 1H, 1×CH_ar_); 7.81–7.86 (m, 1H, 1×CH_ar_); 8.07 (s, 1H, 1×CH_ar_); 9.70–9.80 (m, 3H, 3×NH); 12.40 (s, 1H, 1×NH). Anal. Calcd. for C_14_H_15_N_5_O_2_S (317.37): C 52.98; H 4.76; N 22.07. Found C 53.08; H 4.75; N 22.00.

*4-(3,4-Dimethoxyphenyl)-1-(4-methylimidazol-5-oyl)thiosemicarbazide* (**13**)**.** Yield: 81%. M.p. 223–225 °C. ^1^H-NMR (DMSO-d_6_) δ (ppm): 2.45 (s, 3H, CH_3_); 3.72 (s, 3H, OCH_3_); 3.73 (s, 3H, OCH_3_), 6.87–6.90 (m, 1H, 1×CH_ar_); 6.96–6.99 (m, 1H, 1×CH_ar_); 7.15 (s, 1H, 1×CH_ar_); 7.62 (s, 1H, 1×CH_ar_); 9.45–9.50 (m, 2H, 2×NH); 9.70, 12.40 (2s, 2H, 2×NH). Anal. Calcd. for C_14_H_17_N_5_O_3_S (335.38): C 50.14; H 5.11; N 20.88. Found C 50.19; H 5.12; N 20.80.

*4-(3-Chloro-4-methylphenyl)-1-(4-methylimidazol-5-oyl)thiosemicarbazide* (**14**)**.** Yield: 89%. M.p. 213–215 °C. ^1^H-NMR (DMSO-d_6_) δ (ppm): 2.29 (s, 3H, CH_3_); 2.45 (s, 3H, CH_3_); 7.25–7.28 (m, 1H, 1×CH_ar_); 7.35–7.38 (m, 1H, 1×CH_ar_); 7.62 (s, 1H, 1×CH_ar_); 7.64 (s, 1H, 1×CH_ar_); 9.64–9.76 (m, 3H, 3×NH); 12.40 (s, 1H, 1×NH). Anal. Calcd. for C_13_H_14_ClN_5_OS (323.80): C 48.22; H 4.36; N 21.63. Found C 48.06; H 4.37; N 21.65.

*4-(2,5-Difluorophenyl)-1-(4-methylimidazol-5-oyl)thiosemicarbazide* (**15**)**.** Yield: 79%. M.p. 218–220 °C. ^1^H-NMR (DMSO-d_6_) δ (ppm): 2.43 (s, 3H, CH_3_); 7.57–7.69 (m, 3H, 3×CH_ar_); 9.35 (s, 1H, 1×CH_ar_); 10.70, 12.41, 13.74 (3s, 4H, 4×NH). Anal. Calcd. for C_12_H_11_F_2_N_5_OS (311.31): C 46.30; H 3.56; N 22.50. Found C 46.45; H 3.57; N 22.54.

*1-(4-Methylimidazol-5-oyl)-4-(2-methylphenyl)thiosemicarbazide* (**16**)**.** Yield: 96%. M.p. 210–212 °C. ^1^H-NMR (DMSO-d_6_) δ (ppm): 2.40 (s, 3H, CH_3_); 2.43 (s, 3H, CH_3_); 7.56–7.64 (m, 4H, 4×CH_ar_); 9.34 (s, 1H, 1×CH_ar_); 10.70, 12.35, 13.73 (3s, 4H, 4×NH). Anal. Calcd for C_13_H_15_N_5_OS (289.36): C 53.96; H 5.23; N 24.20. Found C 53.76; H 5.22; N 24.17.

*1-(4-Methylimidazol-5-oyl)-4-(3-methylphenyl)thiosemicarbazide* (**17**)**.** Yield: 86%. M.p. 193–195 °C. ^1^H-NMR (DMSO-d_6_) δ (ppm): 2.23 (s, 3H, CH_3_); 2.45 (s, 3H, CH_3_); 6.92–6.95 (m, 1H, 1×CH_ar_); 7.16–7.21 (m, 1H, 1×CH_ar_); 7.30–7.34 (m, 2H, 2×CH_ar_); 9.50–9.75 (m, 3H, 3×NH); 12.40 (s, 1H, 1×NH). Anal. Calcd. for C_13_H_15_N_5_OS (289.36): C 53.96; H 5.23; N 24.20. Found C 54.01; H 5.24; N 24.27.

*1-(4-Methylimidazol-5-oyl)-4-(4-dimethyloaminophenyl)thiosemicarbazide* (**19**)**.** Yield: 94%. M.p. 215–217 °C. ^1^H-NMR (DMSO-d_6_) δ (ppm): 2.45 (s, 3H, CH_3_); 2.87 (s, 6H, 2×CH_3_); 6.65–6.70 (m, 2H, 2×CH_ar_); 7.16–7.22 (m, 2H, 2×CH_ar_); 7.61 (s, 1H, 1×CH_ar_); 9.15, 9.32, 9.69, 12.39 (4s, 4H, 4×NH). Anal. Calcd. for C_14_H_18_N_6_OS (318.40): C 52.81; H 5.70; N 26.39. Found C 52.94; H 5.69; N 26.47.

*1-(4-Methylimidazol-5-oyl)-4-(4-diethyloaminophenyl)thiosemicarbazide* (**20**)**.** Yield: 81%. M.p. 225–227 °C. ^1^H-NMR (DMSO-d_6_) δ (ppm): 1.05–1.10 (t, 6H, 2×CH_3_); 2.45 (s, 3H, CH_3_); 3.28–3.32 (m, 4H, 2×CH_2_); 6.58-6.61 (dd, 2H, 2×CH_ar_); 7.14–7.17 (dd, 2H, 2×CH_ar_); 7.61 (s, 1H, 1×CH_ar_); 9.27, 9.36, 9.68, 12.38 (4s, 4H, 4×NH). Anal. Calcd. for C_16_H_22_N_6_OS (346.45): C 55.47; H 6.40; N 24.26. Found C 55.57; H 6.38; N 24.18.

*4-(4-Ethoxyphenyl)-1-(4-methylimidazol-5-oyl)thiosemicarbazide* (**21**)**.** Yield: 84%. M.p. 210–212 °C. ^1^H-NMR (DMSO-d_6_) δ (ppm): 1.29–1.34 (t, 3H, CH_3_); 2.45 (s, 3H, CH_3_); 3.96–4.02 (q, 2H, 1×CH_2_); 6.84–6.87 (dd, 2H, 2×CH_ar_); 7.30–7.33 (dd, 2H, 2×CH_ar_); 7.61 (s, 1H, 1×CH_ar_); 9.44, 9.72, 12.38 (3s, 4H, 4×NH). Anal. Calcd. for C_14_H_17_N_5_O_2_S (319.38): C 52.65; H 5.37; N 21.93. Found C 52.52; H 5.38; N 22.00.

*4-(2,6-Diisopropylphenyl)-1-(4-methylimidazol-5-oyl)thiosemicarbazide* (**22**)**.** Yield: 73%. M.p. 202–204 °C. ^1^H-NMR (DMSO-d_6_) δ (ppm): 1.29–1.31 (d, 12H, 4×CH_3_); 2.43 (s, 3H, CH_3_); 3.21–3.27 (sept., 2H, 2×CH_isopropyl_); 7.05–7.34 (m, 3H, 3×CH_ar_); 7.60 (s, 1H, 1×CH_ar_); 9.00, 9.33, 9.42, 12.34 (4s, 4H, 4×NH). Anal. Calcd. for C_18_H_25_N_5_OS (359.49): C 60.14; H 7.01; N 19.48. Found C 60.01; H 7.02; N 19.54.

### 3.3. Cytotoxicity Assays

#### 3.3.1. Cell Culture

Cell line L929 mouse fibroblast (ATTC^®^ CCL-1™) was routinely cultured in IMDM (Iscove’s Modified Dulbecco Medium—Biowest, Nuaille, France) supplemented with 10% HIFBS (Heat-Inactivated (1 h in 56 °C) Fetal Bovine Serum—Biowest), 100 U/mL penicillin and 100 μg/mL streptomycin. Cells were trypsinized twice a week, and seeded at density 1 × 10^6^ per T25 cell culture flask, and incubated for 24–48 h at 37 °C and 10% CO_2_ to achieve a confluent monolayer.

#### 3.3.2. Preparation of Compounds and Sulfadiazine

Compounds **2**–**22** were dissolved in dimethyl sulfoxide (DMSO—Sigma) to 50,000 μg/mL. The final concentration of DMSO in compounds dilutions was not higher than 1.00%. Sulfadiazine [4-amino-*N*-(2-pyrimidinyl)benzenesulfonamide] (S8626, Sigma) was dissolved in 1 M sodium hydroxide (NaOH—Sigma) to 100 mg/mL. Final concertation of NaOH in sulfadiazine dilutions was not higher than 2.5%. Dilution of the all compounds **2**–**22** and drug were freshly prepared before the cells were exposed.

#### 3.3.3. Cell Viability Assay

The effects of tested compounds and drugs on the viability of mouse fibroblasts L929 cells were evaluated using the MTT salt [3-(4,5-dimethylthiazol-2-yl)-2,5-diphenyltetrazolium bromide] (Sigma). The MTT assay was used according to international standards: ISO 10993-5:2009(E), Biological evaluation of medical devices, Part 5: Tests for *in vitro* cytotoxicity. L929 cells were placed into 96-well plates at a density of 1 × 10^4^/100 μL/well in culture medium RPMI 1640 without phenol red (Biowest) supplemented with 10% HIFBS, 2 mM l-glutamine (Sigma), 100 U/mL penicillin, 100 μg/mL streptomycin, grown at 37 °C in a 10% CO_2_ humidified environment and allowed to attach and form a confluent monolayer for 24 h before treatment. Afterward, the culture medium in the plates was replaced by 100 μL of compounds and sulfadiazine suspension and the cells were exposed for 24 h. Then, 1 mg/mL of MTT solution in RPMI 1640 without phenol red was prepared and 50 μL of the solution was added to each well and incubated at 37 °C, 10% CO_2_ for 2 h. Mitochondrial dehydrogenases of viable cells reduced the yellowish water-soluble MTT to water-insoluble formazan crystals, which were solubilized with dimethyl sulfoxide (DMSO). The cell culture medium was aspirated cautiously, after which 150 μL DMSO was added to each well, the plates were gently mixed, then 25 μL 0.1 M glycine buffer (pH = 10.5) (Sigma) was added. Optical density was read on the ELISA reader (Multiskan EX, Labsystems, Vienna, VA, USA) at 570 nm. The results were expressed as the percentage of viability compared to untreated cells. All experiments were performed in triplicate. Moreover, cells were treated with 4.0–0.03% concentration of DMSO as a compounds solvent (data not shown).

### 3.4. In Vitro Assay for Anti-T. gondii Activity

#### 3.4.1. Cell Line and Parasite Culture

Vero cell line (African green monkey kidney) (ATCC^®^ CCL-81™) were maintained in EMEM (EMEM/10%HIFBS/P/S) (Eagle’s Minimum Essential Medium—ATCC^®^ 30-2003™) culture medium supplemented with 10% HIFBS (Heat-Inactivated (1 h in 56°C) Fetal Bovine Serum—ATCC), 100 U/mL penicillin, 100 μg/mL streptomycin. The cell line was trypsinized (using Trypsin-EDTA Solution, 1X ATCC^®^ 30-2101™) twice a week, and seeded at density 1 × 10^6^ per T25 cell culture flask (Falcon), and incubated for 24–48 h at 37 °C and 5 CO_2_ to achieve a confluent monolayer. The highly virulent RH strain of *Toxoplasma gondii*, haplogroup I (ATCC^®^ PRA-310™) were maintained as tachyzoites according to the ATCC product sheet, in DMEM (Dulbecco’s Modified Eagle’s Medium—ATCC^®^ 30-2002™) (DMEM/3%HIFBS/P/S—parasite culture medium) under condition 37 °C and 5 CO_2_ and passage twice per week.

#### 3.4.2. Influence of 2–22 and Sulfadiazine on *T. gondii* Proliferation

The Vero cells were seeded on 96-well plates (1 × 10^5^ cells/100 µL/well) in EMEM/10%HIFBS/P/S. After 24 h of incubation medium was removed and then *T. gondii* RH tachyzoites were added to the cell monolayers at density 1 × 10^5^ cells/100 µL/well in DMEM/3%HIFBS/P/S. One hour later different compounds and sulfadiazine dilutions (100 µL/well) in parasite culture medium were added to the cell monolayers with *T. gondii*. After subsequent 24 h of incubation 1 µCi/well [5,6-^3^H] uracil (Moravek Biochemicals Inc., Brea, CA, USA) was applied to each microculture for further 72 h. The amount of the isotope incorporated into the parasite nucleic acid pool, corresponding to the parasite growth, was measured by liquid scintillation counting using 1450 Microbeta Plus Liquid Scintillation Counter (Wallac Oy, Turku, Finland). The results were expressed as counts per minute (CPM) and transformed to the percentage of viability compared to untreated cells. All experiments were performed in triplicate.

### 3.5. Graphs and Statistical Analyses

Statistical analyses and graphs were performed using GraphPad Prism version 8.0.1 for macOS (GraphPad Software, San Diego, CA, USA). Additionally, to the established relationship between cytotoxicity and antiparasitic activity typical selectivity indexing is calculated as the CC_30_ or even CC_50_ over IC_50_ values. In this paper we also established this dependence, to prioritize imidazole-thiosemicarbazides as therapeutic agents, through a modified selectivity index, which was calculated as the logarithm of the ratio of the CC_30_ and the IC_50_ values, according to the formula: SI = log(CC_30_/IC_50_). A positive value represents more selectivity against *T. gondii* than L929 cells, while negative ones (or close to zero) indicate higher toxicity to L929 cells and low selectivity to the parasite [46,47]. With a CC_30_ range between 3.91 to 500 µg/mL and IC_50_ values between 1 and 125 µg/mL, the equation gives a SI value between −1.5 to 2.7. It worth underlining that in our study we developed a rigorous approach to determine cytotoxicity by using the CC_30_ instead of CC_50_, which makes our SI value stricter and more precise. The higher the SI value the more selective the imidazole-thiosemicarbazides were towards inhibiting *T. gondii* proliferation vs. cytotoxicity. Therefore, compounds with a positive SI value closer to 2.7 may have potential as agents against toxoplasmosis, while derivatives with a negative SI possesses more toxic properties than specific *T. gondii* in vitro inhibition and received low priority for follow-up. For compounds with IC_50_ values greater than 125 µg/mL, the highest concentration tested, IC_50_ values were calculated based on extrapolation of the curves using GraphPad Prism program (version 8.0.1).

### 3.6. Computational Details

Conformational search was performed using the Amber force field as implemented in HyperChem8.0.3. [48] and default convergence criteria. The rule of five, the number of rotatable bonds, and polar surface area (PSA) were determined using the Molinspiration program [49]. Solubility (log*S*) was calculated using ALOGPS 2.1 program [50]. Absorption (%ABS) was calculated by equation %ABS = 109 − 0.345 × PSA [38].

## 4. Conclusions

The results reported in this paper discusses our prior findings on the inhibitory activity of thiosemicarbazides against *Toxoplasma gondii* proliferation. In particular, we have confirmed that 4-arylthiosemicarbazides with a five-membered heterocyclic ring at the N1 position might be considered as potential anti-*Toxoplasma gondii* agents. The best imidazole-thiosemicarbazides reported here, **3** and **4**, were 125 to 184 fold more potent than sulfadiazine, respectively.

An ideal anti-*Toxoplasma* drug should have efficient tissue penetration, especially through the CNS and the placental barrier. Moreover, should possess activity against tachyzoites related with acute phase, and ability to cyst penetration and bradyzoites inactivation which are responsible for the latent stage of toxoplasmosis. Any drug approved for use also should be free from fetal toxicity and teratogenicity. However, none of the currently used drugs fulfill these criteria. Promising results obtained in this research of thiosemicarbazide derivatives which inhibit the tachyzoites proliferation *in vitro*, at a concentration that is not cytotoxic for the host cells, are the basis for the currently started experiments including *in vivo* tests on a mouse model with another *T. gondii* strain (Me49 and DX) responsible for cyst formation. If found to be effective in our *in vivo* studies, such compounds could represent leads for the development of novel drugs against toxoplasmosis. From the other hand, due to slight differences between tachyzoites and bradyzoites, such as differences in metabolism, morphological changes and stage-specific antigen expression [3,4], we can speculate that our compounds will also have activity against bradyzoites.

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
