# Peer review of "Systematic Identification of Thiosemicarbazides for Inhibition of Toxoplasma gondii Growth In Vitro"

_molecules, 2019, doi:10.3390/molecules24030614_

Reviewer 1 Report

The paper "systemic identification of thiosemicarbazides for inhibition of Toxoplasma gondii growth in vitro" by Paneth A. et al. describes the synthesis of drug derivated from a previous study and the effect on Toxoplasma gondii proliferation.

Overall the paper is well written and fully understandable but lack of discussion.

The paper is of interest for Molecules' readers.

Minor comments:

- Figures 2,3, (4?): change "concetration" to "concentration".

- Figure 4: the figure reported here is the figure 3. Please insert the right figure to allow complete peer reviewing.

- Figure 2 (and 4?): please improve the quality of the picture like the figure 3.

- Figure legend lines 137, 152, 167: Authors should write : "Dose response curves and IC50 values of compounds xxx on T.gondii proliferation.

Line 167: Authors write "+/-SD" but due to the lack of the figure and regarding the figures 2 and 3, it seems not to be shown. Authors must pay attention on that for the revised version.

Line 179: change "standard derivation" to "standard deviation"

Line 313: "Preparation of compounds and:" And what ? Authors must correct.

Line 338: Authors should start by "In vitro assay for ..."

Table 2 and 3: define "Cmpd" or write "compound".

Major Comments :

- Please correct the figure 4.

- In the introduction, authors should deal with other strategy for curing T.gondii such as vaccination, prevention... and eventually focus on that fails to highlight their strategy.

- There is no results nor a real discussion on the activity of the compounds on bradyzoites. Authors should introduce the T.gondii cycle because cysts could reactivate (immunodepressed patients...) and lead to acute toxoplasmosis infection. Fighting tachyzoite AND bradyzoite is the solution. Authors should discuss that point.

- T.gondii cysts are often found in muscles or brain. Authors should discuss how their compounds could target these tissues in vivo... (or better, do ADME assay on the best drug(s)... but it is probably another paper)

Author Response

Response to comments

Detailed responses (marked in blue) to reviewers’ comments

Reviewer 1:

English language and style are fine/minor spell check required 

The manuscript has been edited by an English-speaking native.

Figures 2,3, (4?): change "concetration" to "concentration".

Figures have been improved.

Figure 4: the figure reported here is the figure 3. Please insert the right figure to allow complete peer reviewing.

Right figure has been inserted into the manuscript.

Figure 2 (and 4?): please improve the quality of the picture like the figure 3. 

The quality of the Figures has been improved.

Figure legend lines 137, 152, 167: Authors should write : "Dose response curves and IC50 values of compounds xxx on T.gondii proliferation.

The legends of Figures have been improved as suggested (now line: 172, 190, 209).

Line 167: Authors write "+/-SD" but due to the lack of the figure and regarding the figures 2 and 3, it seems not to be shown. Authors must pay attention on that for the revised version.

Missing Fig. 4 was added. SD values were plotted on graphs shown in Figs. 2-4 (now line: 172, 190, 209).

Line 179: change "standard derivation" to "standard deviation."

It has been corrected. (now line: 222).

Line 313: "Preparation of compounds and:" And what ? Authors must correct.

Now line 363. And sulfadiazine. It has been corrected. (now line: 355).

Line 338: Authors should start by "In vitro assay for ..."

It has been corrected as suggested. (now line: 380).

Table 2 and 3: define "Cmpd" or write "compound".

Tables 2 and 3 have been corrected as suggested.

Please correct the figure 4.

The Figure 4 has been corrected.

Chemical structures of all the derivatives should be provided.

Chemical structures of all the derivatives have been added.

In the introduction, authors should deal with other strategy for curing T.gondii such as vaccination, prevention... and eventually focus on that fails to highlight their strategy.

Appropriate paragraph has been added to the Introduction as suggested (line:88-95).

„Summarizing, currently, the only effective mean of preventing T. gondii infection is a preventive healthcare, especially raising the awareness of future mothers and early diagnosis of pregnant women, and new-borns. An efficient method of complete elimination of the parasite from an infected organism has not yet been developed, so new agents or combinations of agents with greater therapeutic efficacy are necessary. Also, develop of safe and efficient tools for immunoprophylaxis of toxoplasmosis is still needed. Nowadays, only one vaccine containing live attenuated tachyzoites of T. gondii S48 strain, are available, but the potential use of the vaccine is restricted to the veterinary purposes because of the possible reversion of the attenuated mutant to the virulent strain.”

There is no results nor a real discussion on the activity of the compounds on bradyzoites. Authors should introduce the T.gondii cycle because cysts could reactivate (immunodepressed patients...) and lead to acute toxoplasmosis infection. Fighting tachyzoite AND bradyzoite is the solution. Authors should discuss that point.

As suggested by the reviewer, the following information is provided (line:24-53)

“According to the World Health Organization, approximately up to one third of the world’s population is infected with Toxoplasma gondii (T. gondii). This protozoan is an obligate intracellular parasite with a complex life cycle which requires intermediate and definitive hosts. Intermediate hosts are especially all warm-blooded animals including most livestock, and humans. While T. gondii infect intermediate hosts and asexually reproduce in them, the only known definitive hosts in which this parasite may complete life cycle and sexually reproduce are members of the family Felidae. Cats become infected mainly through predation of intermediate hosts with latent parasite invasion. The parasite’s sexual replication takes place in the intestines and results in formation of oocysts. Cats shed large numbers of unsporulated oocysts with feces, for 1-3 weeks. In the environment oocysts sporulate within 1-5 days and become infective.

In T. gondii life cycle we can identify three main infective stages: tachyzoites, tissue cysts with bradyzoites and above-mentioned mature oocyst containing sporozoites. The main rout of parasite transmission to humans involves ingestion of either raw or underprepared meat containing tissue cysts or water or vegetables contaminated with soil containing oocysts. Additionally, people can become infected horizontally (iatrogenic) via blood transfusion or organ transplantation and vertically from mother to fetus via placenta. The parasite is also responsible for livestock infections. Farm animals, also these bred for human consumption, can acquire T. gondii infection through ingestion of sporulated oocysts with water or plants. After the release from tissue cysts and oocysts, which takes place in the intestines, sporozoites and bradyzoites, respectively, transform into the rapidly dividing tachyzoite (tachos – speed in Greek) that is responsible for acute toxoplasmosis. In the immunocompetent individuals tachyzoites under the pressure of immune component convert into slow-dividing bradyzoites (brady – slow in Greek) enclosed within tissue cysts localized in various tissues e.g. neural or/and muscle. In people with a fully effective immune response those tissue cysts do not possess a direct threat to health and even life. In contrast, in immunocompromised patients the rupture of tissue cysts leads to the release of bradyzoites. Their transformation to tachyzoites, in the absence of: nitric oxide, INF-g, TNF-a, T cells and IL-12, results in disease reactivation. The process of tachyzoite–bradyzoite conversion is central to the pathogenesis and longevity of infection. Therefore, the biggest challenge in the treatment of toxoplasmosis is related with acute phase, when rapidly multiplying tachyzoites are responsible for the numerous of necrotic changes and destruction of the host cells.”

T. gondii cysts are often found in muscles or brain. Authors should discuss how their compounds could target these tissues in vivo... (or better, do ADME assay on the best drug(s)... but it is probably another paper)

As suggested by the reviewer, the following information is provided (now line: 433-445).

“An ideal anti-Toxoplasma drug should have efficient tissue penetration, especially through the CNS and the placental barrier. Moreover, should possess activity against tachyzoites related with acute phase, and ability to cyst penetration and bradyzoites inactivation which are responsible for latent stage of toxoplasmosis. Drug approved for use also should be free from fetal toxicity and teratogenicity. However, none of the currently used drugs fulfilled these criteria. Promising results obtained in this research of thiosemicarbazide derivatives which inhibit the tachyzoites proliferation in vitro, in not cytotoxic concentration for the host cells, are the basis for the currently started experiment including in vivo tests on a mouse model with another T. gondii strain (Me49 and DX) responsible for cyst formation. If they are found to be effective in in vivo studies, such compounds could represent leads in the development of novel drugs against toxoplasmosis. From the other hand, due to slight differences between tachyzoite and bradyzoite, such as alterations to metabolism, morphological changes and stage-specific antigen expression, we can speculate that our compounds have also activity against bradyzoites..”

Reviewer 2 Report

The present work presents an excellent quality of experiments and an interesting perspective of new drugs for use in Toxoplasmosis. This area of study has had few concrete results in recent years with few new drugs keeping the same drug for many years. The authors have shown interesting and promising results.

Author Response

Moderate English changes required

The manuscript has been edited by an English-speaking native.